# Is Temporal-Difference Learning the Only Path to Stitching in RL?

## Abstract

Reinforcement learning (RL) promises to solve long-horizon tasks even when training data contains only short fragments of the behaviors. This *experience stitching* capability is often viewed as the purview of temporal difference (TD) methods. However, outside of small tabular settings, trajectories never intersect, calling into question this conventional wisdom. Moreover, the common belief is that Monte Carlo (MC) methods should not be able to recombine experience, yet it remains unclear whether function approximation could result in a form of implicit stitching. The goal of this paper is to empirically study whether the conventional wisdom about stitching actually holds in settings where function approximation is used. We empirically demonstrate that Monte Carlo (MC) methods can also achieve experience stitching. While TD methods do achieve slightly stronger capabilities than MC methods (in line with conventional wisdom), this gap narrows as we use larger neural networks. Furthermore, we find that increasing critic capacity effectively reduces the generalization gap for both the MC and TD methods. These results suggest that the traditional TD inductive bias for stitching may be less necessary in the era of large models for RL and, in some cases, may offer diminishing returns. Additionally, our results suggest that stitching, a form of generalization unique to the RL setting, might be achieved not through specialized algorithms (temporal difference learning) but rather through the same recipe that has provided generalization in other machine learning settings (via scale). Project website: https://anonymous.4open.science/r/a-broken-promise-F5FB/README.md

## 1 Introduction

In theory, reinforcement learning algorithms should be able to piece together past experiences to find new or better solutions to long-horizon tasks. This ability, sometimes called *experience stitching* (Ghugare et al., 2024; Myers et al., 2025; Wolczyk et al., 2024; Ziebart et al., 2008), is frequently linked to bootstrapping through temporal-difference (TD) updates, i.e., updating value estimates using successor states' predictions instead of relying on full rollouts. At least in tabular settings, TD-based methods can boost data efficiency and accelerate convergence (Sutton, 1988; Sutton & Barto, 2018), yet their efficacy in the presence of function approximation remains disputed (Bertsekas, 1995; 2010; Brandfonbrener et al., 2021; Peters et al., 2010).

Outside of tabular or highly-constrained settings, TD methods cannot literally stitch trajectories together: trajectories rarely self-intersect in real-world scenarios. For example, compare an ant crawling on a sheet of paper (2D) with a fly flying in an empty room (3D) in Figure 1: the ant's 2D path will self-cross far more often than the

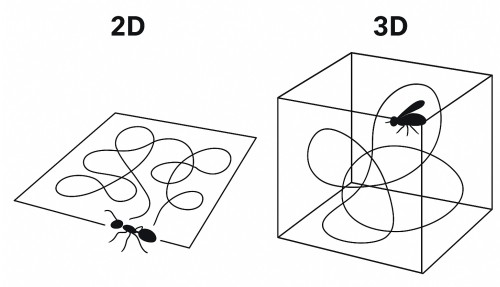

**2D**      **3D**

Figure 1: *(Left)* While TD methods are often conceptualized as piecing together overlapping trajectories, *(Right)* this mental model breaks down in almost all realistic tasks, as trajectories never actually intersect. This paper introduces a new mental model (and formal definitions) for thinking about "stitching" in such settings, provides a benchmark for rigorously evaluating these stitching capabilities, and performs experiments to understand the degree to which stitching may *actually* achieved through *(i)* temporal difference methods, *(ii)* quasimetric architectures, and *(iii)* simply scaling model architectures.

fly's 3D path. Following this example, we observe that stitching has a dual relationship with generalization. On one hand, stitching requires generalization: the value function must assign similar values to similar states, enabling values to propagate across disconnected trajectories. On the other hand, stitching itself provides generalization: it allows a policy to traverse between states that were never observed as connected during training.

In this paper, we examine mechanisms that enable recombining high-dimensional experiences. We focus on model scale and learning paradigm (TD vs. MC), and we evaluate three regimes—*no stitching*, *exact stitching* (shared waypoint), and *generalized stitching* (waypoint mismatch). Here, a waypoint refers to an intermediate state along the trajectory. To probe them cleanly, we introduce a minimalist pick-and-place grid benchmark (Sokoban without walls; but with lift/drop actions) designed to test composition rather than perception or complex dynamics. Two setups anchor our study: Quarters (exact stitching), where training transfers boxes between adjacent board quarters and evaluation requires a diagonal transfer; and Few-to-Many (generalized stitching), where training solves easier instances with some boxes pre-placed while evaluation requires moving all boxes. We distinguish *closed* stitching cases—where composed solutions remain within the support of the training data—from *open* cases—where they typically fall outside; formal definitions appear in Section 4.

The main contribution of this paper is a carefully designed testbed and empirical evaluation of the stitching capabilities of various algorithms and architectures. We train 7 different goal-conditioned agents in this environment and observe that **Monte Carlo methods do stitch**: in the generalized regime, they achieve small generalization gaps—often comparable to TD—even when training requires moving fewer objects than at evaluation. At the same time, **exact stitching with multi-object coordination is brittle**: performance degrades rapidly as the number of objects grows, and even TD can fail when composition steers rollouts through intermediate states that were never seen during training. In addition, we find that **scale is a powerful lever for stitching**. Increasing the size of the critic network, used for state-action pair value estimation, substantially boosts test performance for both TD and MC variants, narrowing their gap; among MC baselines, algorithms with stronger exploration and credit assignment fare best, while lightweight MC DQN lags primarily due to exploration inefficiency. Taken together, these results revise common wisdom: TD is neither necessary for stitching, nor sufficient in the face of multi-object composition; model scale materially improves stitching for both paradigms.

Our main contributions are the following:

1. We formalize and analyze three stitching regimes—*no stitching*, *exact stitching* (shared waypoint), and *generalized stitching* (waypoint mismatch)—and highlight when exact-stitching evaluations can break due to lack of closure under composition.

2. Through controlled experiments across TD and MC algorithms, we provide principled guidance on stitching: MC methods can stitch in the generalized regime, TD typically helps but is insufficient, and increasing critic scale markedly improves stitching for both paradigms.

3. We introduce simple, configurable environments that isolate stitching phenomena and enable reproducible evaluation across regimes (see Fig. 3).

## 2 RELATED WORK

**From tabular prediction to stitching.** Early reinforcement learning emphasized value estimation in tabular models, grounded in dynamic programming (Bellman & Kalaba, 1957). TD learning realizes this idea via bootstrapping from successor predictions (Sutton, 1988; Sutton & Barto, 2018), with extensions beyond the tabular regime through residual-gradient, least-squares TD, and linear-convergence analyses (Baird et al., 1995; Bradtke & Barto, 1996; Tsitsiklis & Van Roy, 2002; Bertsekas & Tsitsiklis, 1995). A natural next step is *generalization across state–goal pairs*, i.e., solving new *combinations* of familiar states and goals—what many works refer to as *stitching* (e.g., UVFA, HER, and successor-feature routes to recomposition) (Kaelbling, 1993; Schaul et al., 2015; Andrychowicz et al., 2017; Barreto et al., 2017; 2018). We describe stitching regimes (Figure 2) purely by what is present in the replay buffer $\mathcal{D}$ at train time and what is queried at test time.

1. **No stitching (end-to-end only).** *Train:* $\mathcal{D}$ contains end-to-end trajectories $(s' \to g')$. *Test:* evaluate a held-out end-to-end pair $(s \to g)$ (same generator, disjoint pairs) (Sutton & Barto, 2018; Ghosh et al., 2019).

2. **Exact stitching (shared waypoint).** *Train:* $\mathcal{D}$ contains trajectories $(s \to w')$ and $(w' \to g)$ for the *same* waypoint $w'$; no $(s \to g)$. *Test:* evaluate the end-to-end query $(s \to g)$. This setting aligns with classic dynamic programming / temporal-difference propagation across a shared waypoint (Bellman & Kalaba, 1957; Sutton, 1988) and recent discussions of "stitching" (Ghugare et al., 2024).

3. **Generalized stitching (waypoint mismatch).** *Train:* $\mathcal{D}$ contains $(s \to w')$ and $(w'' \to g)$ with $w' \neq w''$; there is no waypoint $\tilde{w}$ for which both trajectories $(s \to \tilde{w})$ and $(\tilde{w} \to g)$ are present. *Test:* evaluate $(s \to g)$. Success requires a representation that bridges mismatched trajectories (e.g., successor features with GPI, temporal distance/value models) (Barreto et al., 2017; 2018; Pong et al., 2018; Ghugare et al., 2024).

**Compositional generalization and horizon extension.** Generalization fragility in deep RL has been documented under controlled shifts in observations, dynamics, and tasks (Zhang et al., 2018; Packer et al., 2019; Cobbe et al., 2020). A complementary lens is *horizon generalization*, where agents trained on short-range goals succeed at longer-range ones by composing waypoints; recent

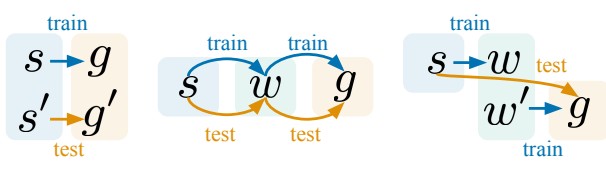

A. No Stitching    B. Exact Stitching    C. Generalized Stitching

Figure 2: Three types of stitching.

work formalizes links to planning invariances and proposes diagnostics (Myers et al., 2025). Parallel lines in ML study *compositional* generalization as systematic recombination of known primitives (e.g., SCAN, CFQ), clarifying what kinds of recomposition are actually measured (Lake & Baroni, 2018; Keysers et al., 2020; Hupkes et al., 2020). Complementary operator-centric approaches propose alternatives to Bellman backups that directly encode subgoal composition to accelerate value propagation in goal-reaching MDPs (Piekos et al., 2023; Van Niekerk et al., 2019; Adamczyk et al., 2023). We adopt this compositional lens and ask whether agents can solve novel state–goal combinations by recombining familiar parts to solve longer tasks.

**Goal-conditioned RL (GCRL) and representation routes to recomposition.** Goal conditioning makes recomposition operational by training policies or value functions over $(s, g)$ pairs. UVFA amortize structure-sharing across goals (Schaul et al., 2015), while HER densifies sparse reward learning by relabeling achieved goals (Andrychowicz et al., 2017). Supervised-learning formulations such as GCSL trade bootstrapping for stability and simplicity (Ghosh et al., 2019), though analyses suggest they may lack stitching without explicit temporal augmentation (Ghugare et al., 2024). Beyond standard backups, the *Compositional Optimality Equation* (COE) replaces Bellman's max-over-actions with an explicit composition over intermediate subgoals, yielding more efficient value propagation in deterministic goal-reaching settings (Piekos et al., 2023). Representation-centric methods also support recomposition via factorization or predictive structure: successor features with generalized policy improvement transfer across reward mixtures (Barreto et al., 2017; 2018), and temporal-difference models learn goal-conditioned distances that enable waypointing and short-horizon planning (Pong et al., 2018; Nasiriany et al., 2019). Building on these strands, we contrast TD-style and MC/SL-style training while varying model capacity to examine whether stitching stems from bootstrapping, from learned representations, or from operator design.

**Stitching in offline RL and explicit trajectory recomposition.** Recent offline RL work makes *trajectory stitching* explicit by learning or constructing joins between sub-trajectories to improve policies from imperfect datasets (Char et al., 2022; Hepburn & Montana, 2022; Li et al., 2024; Ghugare et al., 2024). This stands in contrast to the *implicit* composition often attributed to TD-style value propagation. A natural question, then, is: *which ingredients are actually needed for stitching to emerge in the **online**, goal-conditioned setting?* We investigate this in a controlled online benchmark where (i) the availability of reusable segments and (ii) whether their concatenation stays on-support (closed) or induces off-support states (open) are both tunable by design.

**Planning-heavy testbeds: Sokoban and variants.** Sokoban and Boxoban stress long-horizon reasoning with irreversible moves and maze-like dead ends, where incidental trajectory intersections are rare and naive stitching is difficult; hybrid agents that leverage learned rollouts (I2A) and recurrent agents with emergent plan-like computation achieve strong results in these domains (Weber et al., 2017; Guez et al., 2019; Taufeeque et al., 2024). We take inspiration from Sokoban but deliberately remove maze-induced confounds by studying an *open-grid* environment with boxes and targets. The agent can *pick up* (not only push) boxes, eliminating dead ends and allowing us to manipulate the number and placement of boxes across consecutive episodes so that the set of seen goals is precisely controlled. This setup allows us to directly test whether agents stitch together familiar subgoals to solve novel state–goal combinations.

## 3 PRELIMINARIES

Our paper investigates the generalization properties of on-policy goal-conditioned reinforcement learning, focusing on how Temporal Difference and Monte Carlo methods, as well as network architectures for function approximation, influence stitching capabilities.

We study the problem of goal-conditioned reinforcement learning in a deterministic controlled Markov process with states $s \in \mathcal{S}$, goals $g \in \mathcal{S}$, and actions $a \in \mathcal{A}$. We use an environment with deterministic state transitions and sample the initial states from the distribution $p_0(s_0)$. The Q-function, or critic, is defined as $Q^\pi(s, a) = \mathbb{E}_\pi\left[G_t \mid S_t = s, A_t = a, G_t = g,\right]$, where $G_t = \sum_{k=0}^{T-t} \gamma^k R_{t+k+1}$ is the empirically observed future discounted return with a discount factor $\gamma$. We study both Monte Carlo methods, where $Q$-functions are learned from returns ($Q(s_t, a_t) \leftarrow G_t$) (Sutton & Barto, 2018; Eysenbach et al., 2021), and Temporal Difference methods, where they are learned from bootstrapped targets ($Q(s_t, a_t) \leftarrow r(s_t, a_t) + \gamma Q(s_{t+1}, a_{t+1})$) (Sutton, 1988). Throughout this paper, we sample actions from the Boltzmann (softmax) distribution induced by $Q$, with learnable temperature $\tau$. The replay buffer stores trajectory sequences, but training uses random i.i.d. pairs sampled from those sequences.

## 4 A BENCHMARK FOR STITCHING

To precisely probe these types of stitching, we constructed a benchmark (Fig. 3) where an agent can pick up and place blocks. Our aim was to create tasks that would allow us to isolate the problems related to different types of stitching, while minimizing the impact of environment complexity, dynamics and agents' perceptual capabilities.

Our environment consists of a square grid of fields, with some fields being occupied by *boxes* and *targets*. The task of the agent is to transfer all of the boxes to targets. This setting is thus similar to Sokoban, but differs in *(1)* removing the walls; and *(2)* lifting/dropping boxes instead of pushing boxes, so there is no possibility for the agent to get stuck in an irreversible state. States are discrete, allowing us to determine exactly whether the agent has visited the same state twice. Actions are also discrete, removing policy learning as a potential confounding factor. Nonetheless, the number of states can be made arbitrarily large; for example, Fig. 3 (b) shows 3 blocks in a $4 \times 4$ room, so the total number of block configurations is $\binom{16}{3} = 560$. If we increase the number of blocks to 8, and the grid size to $5 \times 5$, the number of configurations is more than a million.

**Observation and action spaces.** The observation consists of `grid_size` × `grid_size` integers, each representing the total information about one respective field of the grid. The goal observation consists of a grid with boxes placed in desired positions. It is important to note that the target markers are added only for human visibility - they are **not** part of the observations, or the goal observations. There are six possible actions that the model can perform in each state: `go_left`, `go_right`, `go_up`, `go_down`, `pick_up_box`, `put_down_box`.

Within this environment, we constructed three major distributions of box and goal placement to test the exact and generalized stitching variants:

- **No stitching** (cf. Fig. 2 A) – A fixed number of blocks are placed randomly, and the goal is a different random arrangement of blocks. This setting was used primarily to check the implementation of algorithm baselines and compare different hyperparameter choices.

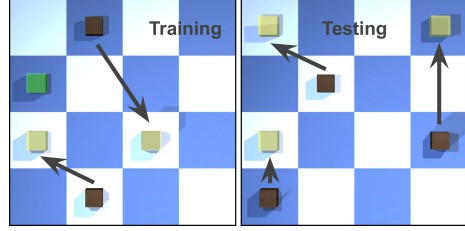

(a) **Exact stitching (Quarters).**      (b) **Generalized stitching (Few-to-Many).**

Figure 3: **A benchmark for stitching:** The agent (red ball) must move boxes to the target positions (yellow transparent boxes). *(Left)* During training, boxes are placed in one quarter and must be moved to an *adjacent* quarter (gray arrow indicates the required direction of transfer). During testing, boxes must be moved to the *diagonal* quarter. The gray arrows illustrate one of the valid two-step routes via adjacent quarters (adjacent → adjacent), which were seen separately during training but never as an end-to-end diagonal move.
*(Right)* During training, one box is already on a target, and the agent must place the remaining two. During testing, no boxes start on targets. Although both start and goal configurations are individually familiar, training never includes segments that involve moving three boxes.

- **The Quarters Setting** (Fig. 3a) – The board is split into four equal quarters. During training, the initial state has all blocks randomly placed within one quarter, and the goal state has the blocks randomly placed in an adjacent quarter.

  The algorithm is then evaluated on the same environment **and** additionally it is evaluated on the same number of boxes and targets, that are placed in diagonal (i.e., not adjacent) quarters. Intuitively, during the training, the agent should learn how to move boxes to a neighboring quarter, and during evaluation, it is tested whether it can stitch the gathered experience to move boxes to the opposite quarter. With a sufficient number of experiences collected, each possible combination of boxes and each possible combination of targets should appear in each of the quarters, which means that during the evaluation, both the initial states and goal states have each been seen before (they are not out of distribution). However, the relative position of boxes in the initial state and goal state has never been observed during training, so the pair $(s, g)$ is out of distribution. Thus, this setting evaluates *exact stitching* (cf. Fig. 2 B).

- **The Few-to-Many Setting** (Fig. 3b) – This setting tests how well the agent can generalize to a task that involves moving a different number of boxes. During training, the environment parameters $n$ and $m$ are fixed (i.e., not randomized). Here, $n$ denotes the total number of boxes and targets, which are placed uniformly on the board, and $m < n$ specifies how many boxes are initially spawned on their targets. Thus, the agent only needs to move the remaining $n - m$ boxes to accomplish the task. During the evaluation, none of the boxes are spawned on the targets. By construction, the initial state and goal state are both in the distribution of states seen during training, yet their combination is (by construction) never seen during training. Since training never included segments starting from the zero-placed start, the $s \to w$ is missing for every $w$ at test, so no waypoint is shared across training trajectories—hence this is *generalized stitching* (cf. Fig. 2 C).

These settings allow us to efficiently test the exact and generalized stitching capabilities and to incrementally change the difficulty of the task by manipulating the size of the grid and the number of boxes.

**Interpreting setups difficulty:** *closed* **vs.** *open* **evaluation.**  The three settings above specify *what segments are seen during training* and *what is queried during testing*. Here, we add an annotation that clarifies what can happen during test-time evaluation.

Let $\mathcal{D}$ be the training replay buffer and let $\mathcal{M} := \{\, s \,:\, s \text{ appears in } \mathcal{D} \,\}$ denote its empirical state support. For a test query $(s, g)$, let $\mathsf{Traj}(s, g)$ be the set of feasible trajectories from $s$ that reach $g$.

**Closed.**  We call $(s, g)$ *closed* if all feasible trajectories $\tau \in \mathsf{Traj}(s, g)$ have states all lying in $\mathcal{M}$ (i.e., no out-of-support waypoint is needed).

**Open.**  We call $(s, g)$ *open* if efficient executions naturally visit states outside $\mathcal{M}$—formally, there exists a near-optimal trajectory $\tilde{\tau} \in \mathsf{Traj}(s, g)$ with some intermediate state $w \notin \mathcal{M}$. In our ex-

periments, we make this observable by reporting the fraction of off-support states visited during evaluation.

*For example:* in our Quarter setting, the agent may leave some boxes placed in the waypoint quadrant, and prematurely drop another toward the goal quadrant, creating a state absent in the training support (see Fig. 5). This is analogous to the challenge in imitation learning wherein out-of-distribution actions can lead to states unseen during training (Ross et al., 2011).

Labeling a setup as *open* does not prevent on-support solutions, nor does it forbid an agent from exploring widely during training. It only indicates that typical efficient executions (including near-optimal ones) are likely to traverse states outside $\mathcal{M}$, given how the training trajectories were collected. If the training rollouts cover essentially the entire relevant state space, the same setting evaluation would be effectively *closed*. Conversely, with finite data, even small deviations can push test rollouts off-support, even for policies that perform well on the training task.

## 5 EXPERIMENTS

The primary goal of our experiments is to understand which types of stitching are performed by TD and MC methods that use a critic with function approximation. We also investigate the role of architecture in stitching, focusing on scaling the critic using Wang et al. (2025) ResNet blocks and parametrization using quasimetric networks from Myers et al. (2025). Our aim is not to propose a new method, but to provide a rigorous evaluation of the stitching capabilities of today's methods. In Section 5.1, we describe the experimental setup, and in the consecutive sections, we aim to answer the following research questions:

- Do any of today's methods do stitch (Section 5.2)?
- Are MC methods performing stitching, or is TD learning necessary (Section 5.3)?
- Does scale improve stitching (Section 5.4)?
- Do quasimetric networks improve stitching (Section 5.5)?

### 5.1 EXPERIMENTAL SETUP

To answer the questions above, we test the exact and generalized stitching capabilities of the Deep Q Networks (DQN) (Mnih et al., 2013), Contrastive Reinforcement Learning (CRL) (Eysenbach et al., 2022), C-learning (Eysenbach et al., 2021), and Implicit Q-Learning (IQL) (Kostrikov et al., 2021). We implement both C-learning and DQN in two versions: MC and TD. While C-learning and CRL are reward-free methods, for DQN, we use a sparse reward of 1 when all of the boxes are in the target position and 0 otherwise. We also use hindsight goal relabeling (Andrychowicz et al., 2017) for DQN with 50% of future states and 50% of random states. In the MC version of DQN and IQL, we use discounted returns for the relabeled goal as targets. To that end, we store experience in a trajectory buffer rather than a standard transition buffer. For each sampled trajectory, we relabel all goals to a future state selected using a geometric distribution. We then compute discounted rewards by propagating them backward through the trajectory. Finally, instead of using a bootstrapped target for the Q-update, we use the discounted cumulative reward computed directly from the replay buffer. In most experiments, we use an MLP with two hidden layers, each containing 256 units, followed by post-activation LayerNorm for the critic. In Section 5.4, we instead adopt the architecture from Wang et al. (2025), which employs ResNet blocks, Swish activations, and pre-activation LayerNorm. In particular, we use two ResNet blocks, each with 4 hidden layers and 1024 units per layer. Note that CRL uses two networks as encoders in the critic. We list all the training details and hyperparameters in Appendix B.

We train all methods using the ADAM optimizer for 5 million update steps, collecting a total of 500 million transitions online. Training alternates between full rollouts, data collection, and network updates. For both data collection and evaluation, we sample actions from the Boltzmann (softmax) distribution defined by $Q$. We do *not* use a separate parameterized policy, as our main focus is on the critic's stitching ability, which could later be distilled into an actor. We tune an additional temperature parameter for all the methods so that the entropy of the Q-induced distribution is close to $\ln(|A|/2) \approx 1.1$. In all of the experiments, we use settings introduced in Section 4, which are implemented as parallelized environments for data collection in JAX (Bradbury et al., 2018). As

a performance metric, we use *success rate*, i.e., the number of attempts finished with all boxes in the target positions. In the majority of the plots, we report the interquartile mean of 10 seeds with stratified bootstrap confidence intervals calculated using Agarwal et al. (2021). We use the term *generalization gap* to name the difference between method performance in the training and evaluation task, which differ in our setups.

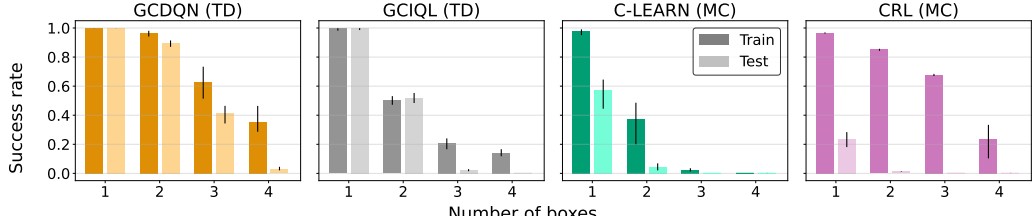

Figure 4: **TD methods can only stitch effectively up to a certain point.** In the Quarters setting (6×6 grid) — which tests exact stitching — increasing the number of boxes widens the generalization gap for both TD and MC methods.

## 5.2 DO ANY OF TODAY'S METHODS DO STITCHING?

Previous works (Ghugare et al., 2024; Myers et al., 2025; Sutton, 1988) argue that temporal difference (TD) methods can compose test-time behavior from sub-behaviors learned during training. However, Monte Carlo (MC) methods might not provide this guarantee. To test this, we probe exact stitching in a Quarter (6×6) grid: can a method recombine learned sub-behaviors to solve a held-out test task? We evaluate one TD method (DQN) and two MC baselines (CRL and C-learn) and report their final performance on both the training and the test tasks.

In Figure 4, DQN (a TD method) achieves near-perfect training and evaluation performance on the single-box task. By contrast, the Monte-Carlo baselines, CRL, and C-learning all show a large generalization gap even in this simplest setting. As the number of boxes increases and test-time observations become out-of-distribution, only DQN retains any nontrivial performance—highlighting an advantage of TD learning. Still, DQN's generalization steadily worsens with more boxes and falls to 0% test performance on the 4-box test task. **A sudden generalization gap of DQN suggests that as the space of possible states expands, it eventually exceeds the stitching capacity of TD updates.** Visual inspection confirms more failures caused by off-support observations with an increased number of boxes (Figure 5). This pattern argues for methods that regularize agent behavior in online RL so agents remain closer to the training observation distribution, analogous to action regularization in offline RL (Fujimoto & Gu, 2021).

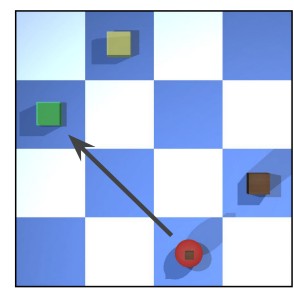

Figure 5: **A subtle failure of stitching.** An agent trained on the quarters task (Fig. 3a) should first move all boxes to an adjacent corner and then to the goal quarter. However, if the agent prematurely moves a box along the diagonal, it will end up in a state that has never been seen before during training.

We also examine generalized stitching in the Few-to-many (5x5) grid, which operates in a closed setup, i.e., all observations are seen in training, and no off-support observations can be visited (Section 4). The test task gets more difficult as the number of boxes spawned at the target position increases, as it demands more stitching. We note that the setting where no boxes are spawned in the target positions corresponds to a no-stitching setup.

We increase the number of boxes spawned at the target position from left to right in Figure 6. For all three baselines, the generalization gap widens as more boxes appear at the target during training. DQN, using TD updates, consistently sustains the highest performance in these harder settings. **Remarkably, CRL still performs well on the test task even when training required moving only three out of four boxes, indicating that an MC-style method can stitch subbehaviors.** We explore this surprising phenomenon in the next sections, using only the Few-to-many setup from now onwards.

## 5.3 ARE MC METHODS PERFORMING STITCHING, OR IS TD LEARNING NECESSARY?

In this section, we compare CRL and TD and MC versions of DQN and C-learning to investigate their stitching capabilities in a generalized closed setup. In particular, we use a Few-to-many 5x5 grid, and train the agent to move 2 boxes to target positions, while a third box is always spawned at the target position. During the test time, all 3 boxes are not in the target position. In Figure 7, we observe that all the methods, except DQN MC, exhibit strong stitching as their generalization gap is relatively small for TD and MC methods, with almost no gap for TD versions of DQN. This result might come as a surprise because MC methods do not employ any explicit mechanisms for stitching, in contrast to TD methods; however, they are still able to work well in this setup, most likely due to implicit stitching on the representation level. The low performance of DQN MC is most probably due to exploration and credit assignment issues.

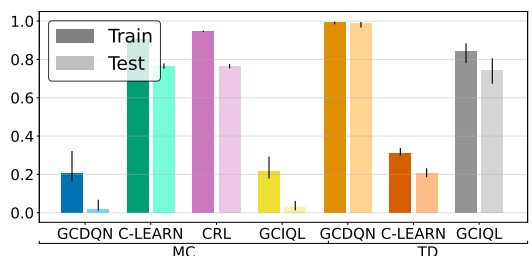

Figure 7: **TD is not necessary for stitching behavior in a generalized setup.** We observe that in the Few-To-Many setting, both TD and MC methods are able to generalize to the test scenario. The exact performance (success rate) varies across different algorithms.

## 5.4 DOES SCALING IMPROVE STITCHING?

Previous works (Nauman et al., 2024; Lee et al., 2025; Wang et al., 2025) have shown that proper scaling of critics' and actors' neural networks can provide enormous benefits in online RL. In this section, we study whether the scale of the critic similarly benefits the stitching capabilities of MC and TD methods. We use the same setup as in Section 5.3. In Figure 8, we show the performance boost on the test task due to using bigger neural networks (extended results are in Figure 10 in Appendix C). CRL, DQN (MC), and C-learn (TD) benefit the most from the critic scale. **Strikingly, the generalization gap might be reduced by simply increasing the scale of the critic for both TD and MC methods.**

Figure 8: **Scale is a powerful lever for stitching.** Increased critic's scale narrows the generalization gap (point distance from the $x = y$ line)—even for MC methods such as CRL.

## 5.5 DO QUASIMETRIC NETWORKS IMPROVE STITCHING?

Quasimetric networks have been shown to provide benefits such as improved sample efficiency in the goal-conditioned RL by making $Q(s, a, g)$ satisfy the triangle inequality (Myers et al., 2025; Liu et al., 2022). In this section, we compare CRL with Contrastive Metric Distillation (CMD) Myers

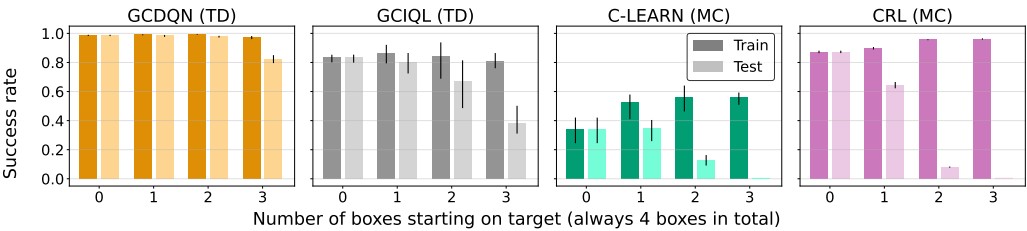

Figure 6: **Stitching is easy on training support, even for MC methods.** In the Few-to-many setting, we probe methods' generalized stitching capabilities. The more difficult the training tasks (fewer boxes starting on target), the smaller the generalization gap for both MC and TD methods.

et al. (2024), which replaces the L2 distance used in CRL with a quasimetric distance between embeddings. We evaluate both methods in the Few-to-many setting on grids 5x5 and 6x6, with 3 boxes. We find that using quasimetric distance only decreases performance and slows down the learning in our benchmark (Figure 9). We believe this is because the environment dynamics is symmetric: for every pair of states A and B, the shortest path from A to B has the same length as from B to A. Thus, splitting embeddings into symmetric and asymmetric components appears to add an unnecessary inductive bias.

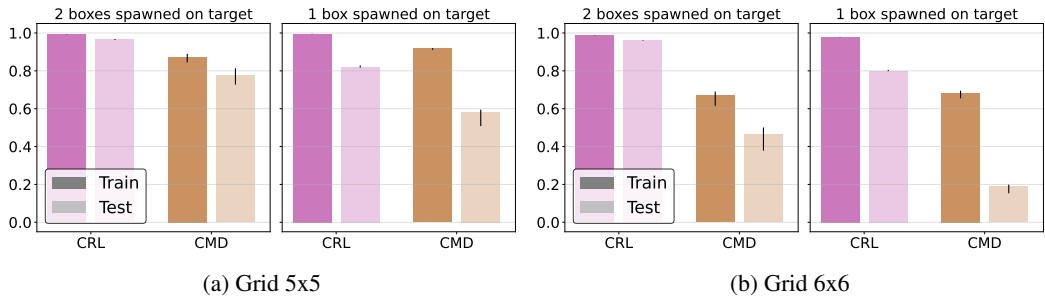

          (a) Grid 5x5                                (b) Grid 6x6

Figure 9: **Quasimetrics do not improve stitching.** In the Few-to-many setting with a 3-box task during testing, CMD results in a worse success rate and a wider generalization gap than CRL.

## 6   Conclusion

This work introduces a formal taxonomy and a controllable benchmark to re-evaluate the mechanisms of experience stitching in goal-conditioned reinforcement learning, yielding key insights that revise conventional wisdom. Our experiments show that, contrary to common belief, Monte Carlo methods can stitch experiences in challenging settings. When test data lies within the training support, they can achieve generalization gaps as small as those of Temporal Difference methods. While TD learning provides an advantage in exact stitching scenarios, its performance degrades as task complexity increases, indicating it is not a universally sufficient solution. Crucially, our results highlight that model scale is a powerful lever for improving stitching. Increasing the critic network's capacity substantially narrows the generalization gap for both MC and TD methods. These findings suggest that the specialized inductive bias of TD learning may be less essential in the era of large models; instead, effective experience stitching can be achieved through the same principle that has proven successful in other machine learning domains: scaling model capacity.

**Limitations.** A key limitation of our work is the reliance on a relatively simple grid-world with a small action space. We chose this controlled setup to enable a concrete evaluation of stitching, which is difficult to verify in richer domains. Nevertheless, even in this simplified setting, temporal-difference methods fail to exhibit exact stitching as the number of boxes increases. Our experiments are further limited to a sparse-reward regime and a small set of popular baselines that we consider representative of goal-conditioned algorithms. We also did not investigate stitching or generalization produced by a separately-parameterized actor policy. Future work should study actor generalization, the effects of exploration and data collection, and scaling to richer, continuous environments.

**Reproducibility Statement.** To ensure the reproducibility of our findings, we provide code and detailed descriptions of our methodology and experimental setup. The repository can be found under the link: `https://anonymous.4open.science/r/a-broken-promise-F5FB/README.md`. The custom grid-world benchmark, including the "Quarters" and "Few-to-Many" settings used to test exact and generalized stitching, is thoroughly described in Section 4 and can be found in `src/envs/block_moving`. Our complete experimental procedure, including the implementation details for all algorithms (DQN, C-Learning, CRL), network architectures, and training protocols, is outlined in Section 5.1 and can be found in `src/impls/agents` and `src/train.py`. Specific hyperparameters used for all experiments, such as learning rates, batch sizes, and discount factors, are enumerated in Table 1 in Appendix B.

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

## A    LLMs USAGE

We used Large Language Models (LLMs) as a writing assistant in the preparation of this manuscript. Their primary role was to aid in restructuring text at both the sentence and paragraph levels to enhance manuscript clarity and readability. Additionally, we used LLMs for proofreading to identify typographical errors and to generate high-level feedback on the paper draft.

## B    EXPERIMENTAL SETUP

All experiments were run on 10 seeds, with the hyperparameters reported in Table 1.

## C    ADDITIONAL RESULTS

### C.1    ARCHITECTURE SCALING

In Figure 8, for readability, we reported the IQM, without confidence intervals. In Figure 10 we present the full scaling experiment results, with confidence intervals, for both grid sizes 4 and 5.

### C.2    HYPERPARAMETER TUNING

To ensure reproducibility and establish a strong baseline, we adopted the core hyperparameter configurations from OGBench (Park et al., 2025) for the IQL (TD) and CRL (MC) implementations. These configurations also served as the starting point for the algorithms implemented specifically for this project: C-learn (MC and TD), DQN (MC and TD), and IQL (MC). However, to verify the suitability of these hyperparameters for the specific challenges of the proposed stitching benchmark, we conducted a sensitivity analysis on critical hyperparameters, including the batch size, number of

Table 1: Hyperparameters

| Hyperparameter | Value |
|---|---|
| num env steps | 500,000,000 |
| num updates | 1,000,000 |
| max replay size (per env instance) | 10,000 |
| min replay size | 1,000 |
| episode length | 100 |
| discount | 0.99 (0.9 for MC versions of DQN, IQL, and C-learning) |
| number of parallel envs | 1024 |
| batch size | 256 |
| learning rate | 3e-4 |
| contrastive loss function | sigmoid_binary_cross_entropy |
| energy function | dot_product |
| representation dimension | 64 |
| target_entropy | 1.1 |

parallel environments, number of gradient updates, discount factor $\gamma$, and the target entropy used in the Q-induced softmax policy. Selected results from this analysis are shown below.

In Figure 11, we report final success rates for different values of discounting and target entropy used in the Q-induced softmax policy during data collection for DQN (TD) and CRL (MC). We use the Quarters Setting with a grid size of $6 \times 6$ and 3 boxes, as this setup yielded results that were neither saturated nor trivial for both DQN and CRL.

We note that while our environment can, in principle, be run with many more boxes and larger grid sizes, all implemented RL methods exhibit relatively low performance in these settings. In practice, they struggle and tend to achieve only trivial performance once the grid size exceeds 6 or the number of boxes reaches four or more.

### C.3 How exploration affects stitching performance?

To study the relationship between data collection entropy (i.e., exploration) and the policy induced by the learned Q function, we conducted experiments with DQN and CRL using different target entropy values in the Q-induced softmax policy during data collection. We use the Quarters Setting with a grid size of $6 \times 6$ and 3 boxes. As shown in Figure 12, the argmax policy for CRL achieves near-zero performance, likely because it gets stuck in states where the Q function is poorly estimated. Policy visualizations suggest that in such cases, the agent either attempts to pick up a box from an empty cell or tries to drop a box it does not have. In contrast, the argmax policy for DQN achieves non-trivial performance, though still lower than that of the softmax policy (see Figure 11(b)), with the best performance occurring at a target entropy of 1.1, the value used in our main experiments.

### C.4 Effects of Discounting and Network Scaling in the Few-to-Many Setup

In this section, we report the effect of the discount parameter on MC and TD methods in the Few-to-many setup, while scaling the architecture of the critic network. We evaluate all methods with discount factor of $\gamma = 0.9$ and $\gamma = 0.99$. Full results of those experiments are presented in Figure 13 and Figure 14. While TD methods have similar performance for both values of $\gamma$, all MC methods but CRL benefit from a lower discount factor, $\gamma = 0.9$. This is likely because lower values of $\gamma$ reduce the variance of relabeled returns (Rathnam et al.; Amit et al., 2020; van Seijen et al., 2019), which can improve generalization, albeit at the cost of shortening the effective horizon.

### C.5 Monte-Carlo DQN results

Most experiments in the paper use 50 training epochs, corresponding to 5 million gradient updates and 500 million environment steps. However, we found that even this substantial amount of training is insufficient for the MC version of DQN to converge when using a small critic architecture and a discount rate of 0.9. In Figure 15, we present the results for a $10\times$ longer training for this method

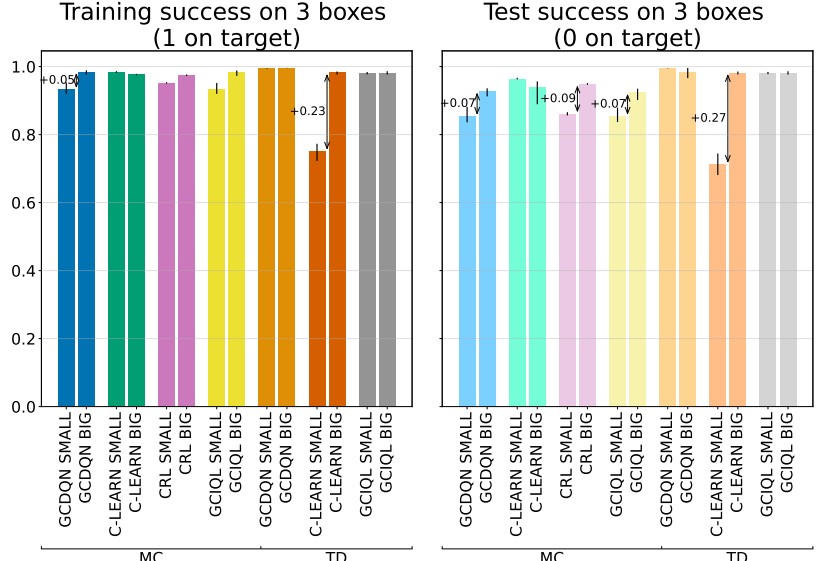

(a) Success rate boost with scaling to bigger networks on grid size of 4 in generalized setup

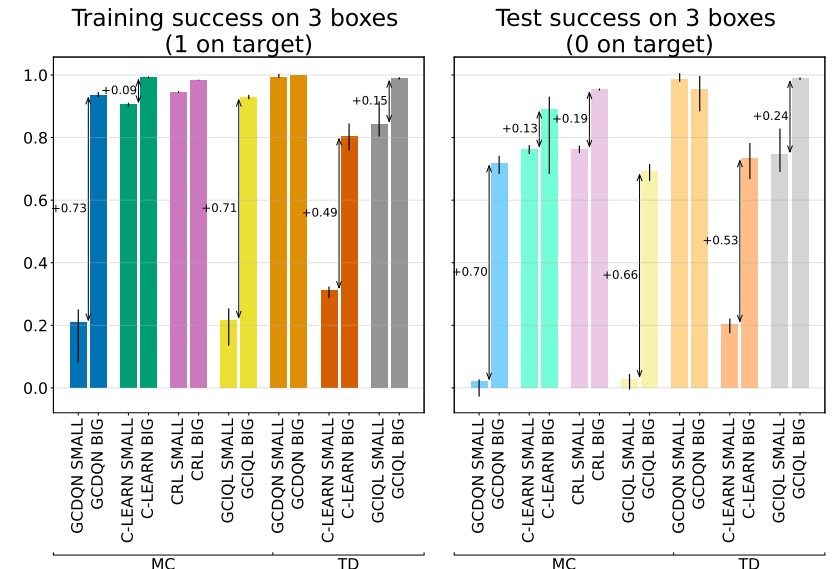

(b) Success rate boost with scaling to bigger networks on grid size of 5 in generalized setup

Figure 10: **Full Generalized Stitching setup experiments**

in the generalized setup (5 × 5) and 4 boxes in total. Interestingly, even when the agent is trained to move only two boxes (green line), it still learns to stitch, achieving a non-trivial success rate of 20% on the test task, which requires moving all four boxes.

## C.6 WALL-CLOCK TIME OF TRAINING

In Table 2, we report the average wall-clock times for training the agents with 500 million environment transitions and 5 million gradient updates, broken down by their architecture sizes. The times are for 5 × 5 grid and the Few-To-Many setup. We conduct experiments using an NVIDIA GeForce GTX 200 120GB GPU. On a single GPU card, we could run up to 5 seeds in parallel.

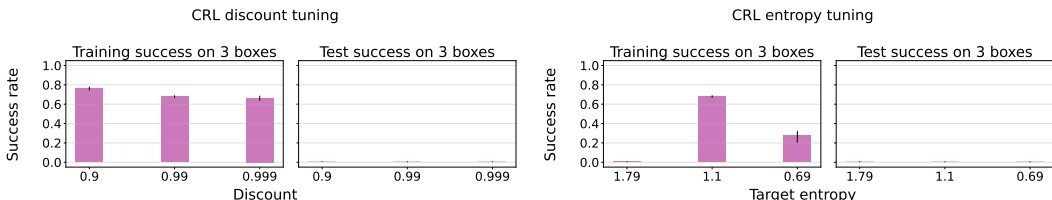

(a) CRL: final train and test success rates for (left) different discounts and (right) different target entropy values used in softmax policy for data collection

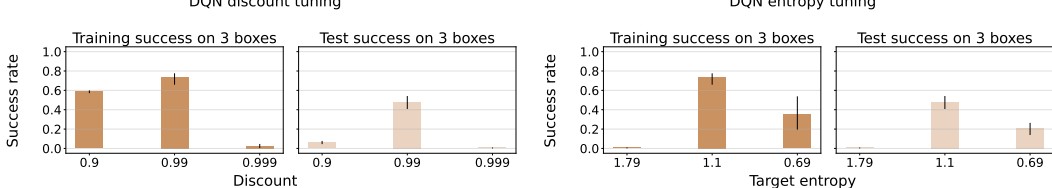

(b) DQN: final train and test success rates for (left) different discounts and (right) different target entropy values used in softmax policy for data collection

Figure 11: **Verification of hyperparameters values.** Success rates for different values of discounting and target entropy used in the Q-induced softmax policy during data collection.

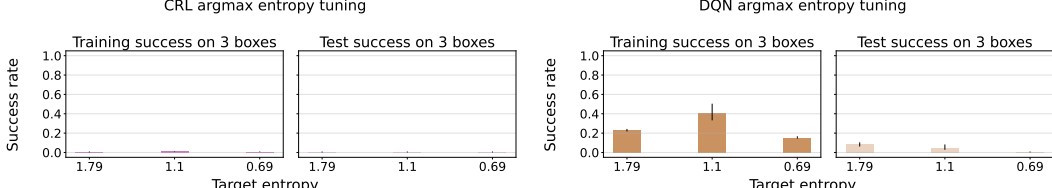

Figure 12: **Relation between data collection policy and argmax policy at test time.** We observe that the argmax(Q) policy yields zero performance for CRL. This suggests that, to prevent CRL from getting stuck in states where the Q-function is misestimated, using a softmax policy at test time is essential. For DQN, the best argmax-policy performance is achieved when the data are collected using a softmax policy with a target entropy of 1.1.

| Depth | DQN (TD) | DQN (MC) | CRL (MC) | IQL (MC) | IQL (TD) | C-LEARN (MC) | C-LEARN (TD) |
|-------|----------|----------|----------|----------|----------|--------------|--------------|
| Small | 1.03 | 1.07 | 1.32 | 1.19 | 1.25 | 0.83 | 1.27 |
| Large | 7.26 | 4.77 | 8.77 | 6.56 | 8.41 | 5.49 | 8.80 |

Table 2: Average wall-clock training time (in hours) for the methods used in this project.

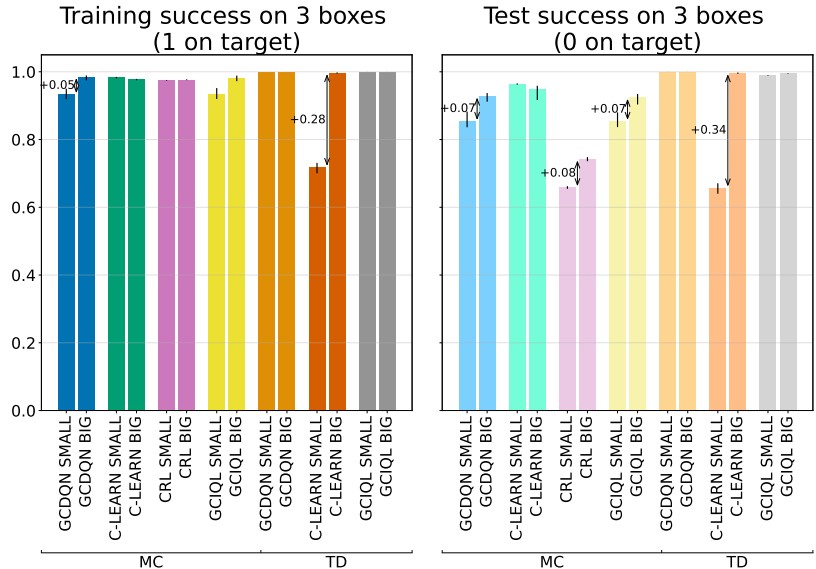

(a) Success rate boost with scaling to bigger networks on grid size of 4 in generalized setup with discount 0.9

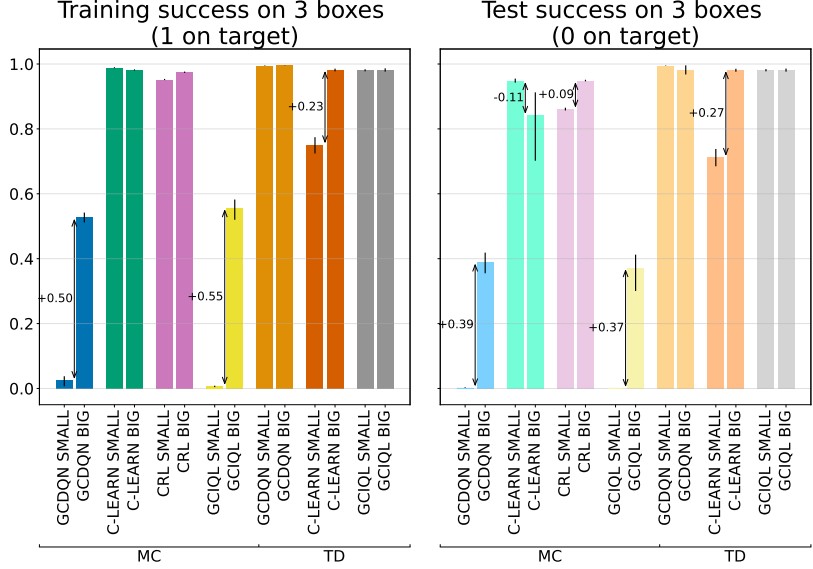

(b) Success rate boost with scaling to bigger networks on grid size of 4 in generalized setup with discount 0.99

Figure 13: **Full discount generalized setup scaling experiments for grid size of 4**

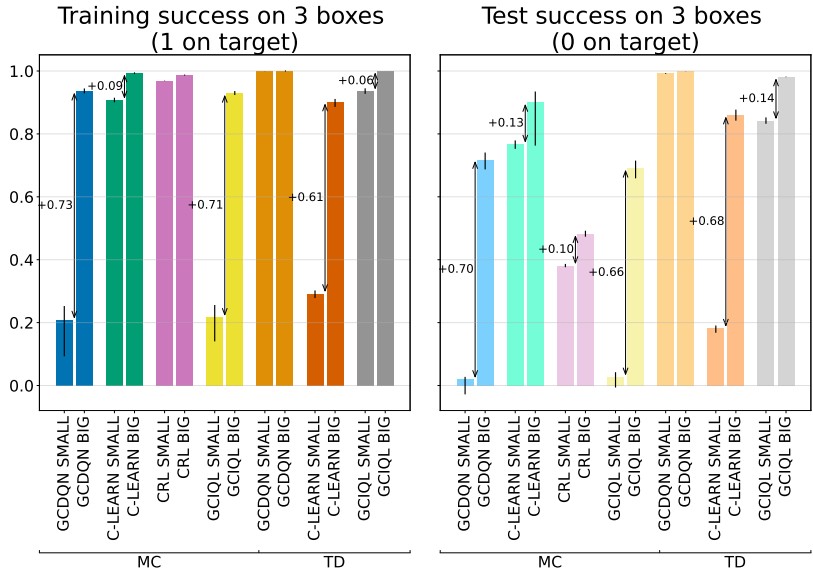

(a) Success rate boost with scaling to bigger networks on grid size of 5 in generalized setup with discount 0.9

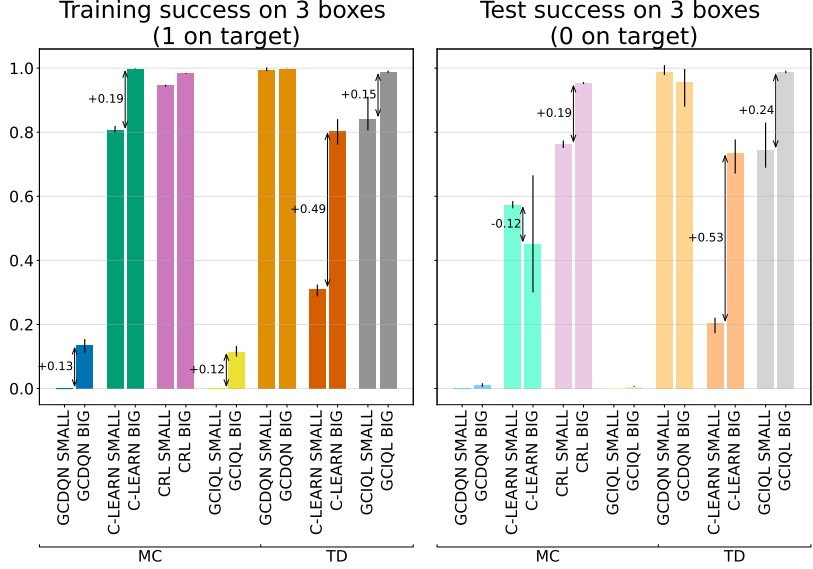

(b) Success rate boost with scaling to bigger networks on grid size of 5 in generalized setup with discount 0.99

Figure 14: **Full discount generalized setup scaling experiments for grid size of 5**

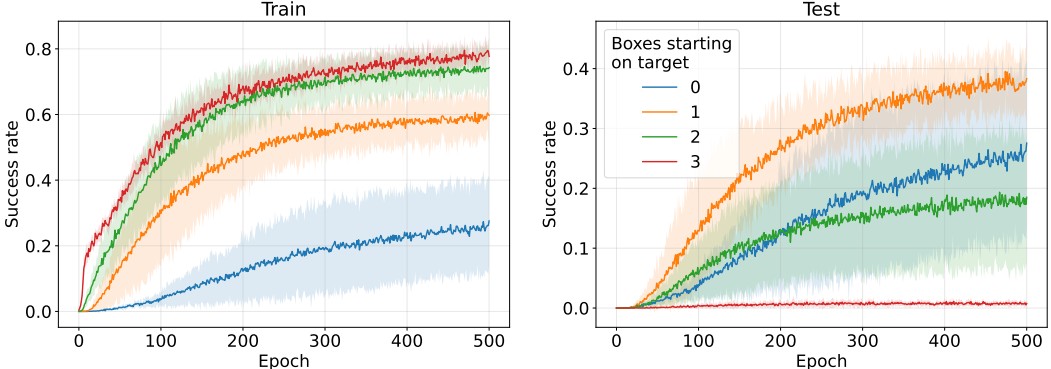

Figure 15: **MC version of Goal-conditioned DQN stitches, but learns slowly.** While the training of MC DQN is slow to converge, the stitching is present for this method even when moving only 3 or 2 boxes out of 4 during the training.

