# OpenReview forum: "Is Temporal Difference Learning the Gold Standard for Stitching in RL?"
_ICLR.cc/2026/Conference — Submitted to ICLR 2026_

### Official Review · Reviewer_18Fp · 2025-10-28

**Soundness:** 2
**Presentation:** 2
**Contribution:** 2
**Rating:** 6
**Confidence:** 2

**Summary:**

The paper revisits a foundational belief in reinforcement learning that temporal difference (TD) learning uniquely enables “experience stitching,” the ability to recombine short trajectory fragments to solve long-horizon tasks. Using carefully controlled goal-conditioned grid environments, the authors design tasks that isolate stitching phenomena across three regimes: no stitching, exact stitching, and generalized stitching. Through extensive experiments comparing TD-based (DQN) and Monte Carlo (MC)-based (C-learning, CRL) methods under different model scales, they find that MC methods can also stitch effectively when model capacity is large, and that scaling the critic reduces generalization gaps more than the TD–MC distinction itself.

**Strengths:**

The study is conceptually insightful and challenges a long-standing assumption in RL with well-designed empirical evidence. The authors introduce a simple yet precise benchmark for studying compositional generalization and provide a clear taxonomy of stitching types. Experimental methodology is strong, consistent architectures, hyperparameters, and evaluation protocols across all baselines ensure fairness and reproducibility. Results are clear: while TD offers a small edge in exact stitching, model scale dominates as the key factor for achieving stitching and generalization. The paper is well-written, balanced, and highly reproducible.

**Weaknesses:**

The work’s main limitation lies in its restricted experimental scope: it focuses on discrete grid-worlds, leaving open whether the same conclusions hold in continuous or visual domains. The paper does not thoroughly analyze exploration efficiency, representation overlap, or the mechanism behind MC stitching. Some theoretical justification for why MC updates might yield implicit compositionality would make the findings more complete. Additionally, the effects of training scale versus network capacity are not fully disentangled, and the runtime or computational costs of scaling are not reported.

**Questions:**

The paper could be strengthened in several ways:
1.	It would be better if the work is extended to more complex domains, such as continuous-control or visual goal-conditioned environments, to test generality beyond discrete grids.
2.	It would be better if the authors include exploration analysis, examining how data diversity or sampling entropy affects stitching success.
3.	It would be better if the paper incorporates actor-critic baselines (e.g., SAC or PPO with goal conditioning) to verify whether the findings hold in standard continuous RL architectures.
4.	It would be better if discussions on theoretical links between representation learning and compositional generalization are given, offering intuition for why scaling may replace traditional TD bootstrapping.

---

> ### Author Response · Authors · 2025-11-21
>
> We thank the reviewer for their time and for reviewing our manuscript. It appears that the reviewer's primary concern is the scope of the presented experiments, which are focused on the grid-world domains.
>
> Weaknesses:
>
> > The work’s main limitation lies in its restricted experimental scope: it focuses on discrete grid-worlds, leaving open whether the same conclusions hold in continuous or visual domains.
>
> While we agree that understanding the stitching introduced by the actor in continuous settings is also important, we pointed out in the limitations section of our work that we chose discrete domains because they offer a controlled setup that enables a concrete evaluation of stitching, which is difficult to study in richer domains. This design choice specifically isolates the stitching capabilities of the critic network, avoiding additional confounding factors such as stitching effects introduced by the actor. That said, stitching induced by the actor is an interesting direction for future work, and we have begun running preliminary experiments on it. If time permits, we will share updates on these results.
>
> > The paper does not thoroughly analyze exploration efficiency, representation overlap, or the mechanism behind MC stitching.
>
> To further investigate how exploration affects stitching performance, we conducted additional experiments during the rebuttal period to examine the effect of the temperature on the softmax Q-induced data collection policy (see Sec. C.3) . Specifically, we monitor the arxgmax(Q) policy on the training and testing task, while using the softmax(Q) policy for data collection with different target temperatures. For both DQN and CRL, using the target entropy of ln(|A|/2) ≈ 1.1 yields the best results in both training and evaluation tasks. Two other values of target entropy: ln(|A|/3) ≈ 0.69 and ln(|A|/1) ≈ 1.79 result in significantly lower argmax(Q) performance, indicating that the target entropy of ln(|A|/2) is a good choice for data exploration.
>
> > Some theoretical justification for why MC updates might yield implicit compositionality would make the findings more complete. Additionally, the effects of training scale versus network capacity are not fully disentangled, and the runtime or computational costs of scaling are not reported.
>
> Although our work is primarily empirical, we hypothesize that increasing model scale enhances stitching by improving alignment of state representations for both MC and TD methods. A deeper investigation of this hypothesis is left for future work.

---

> > ### Author Response · Authors · 2025-11-21
> >
> > Questions
> >
> > > The paper could be strengthened in several ways: 1. It would be better if the work is extended to more complex domains, such as continuous-control or visual goal-conditioned environments, to test generality beyond discrete grids.
> >
> > Please refer to our experiments with PPO in BuilderBench [1] described above.
> >
> > > 2. It would be better if the authors include exploration analysis, examining how data diversity or sampling entropy affects stitching success.
> >
> > Please refer to our experiments with argmax(Q) policy and different target entropies for data collection policies described above.
> >
> > > 3. It would be better if the paper incorporates actor-critic baselines (e.g., SAC or PPO with goal conditioning) to verify whether the findings hold in standard continuous RL architectures.
> >
> > If time permits, we will share updates on these experiments.
> >
> > > 4. It would be better if discussions on theoretical links between representation learning and compositional generalization are given, offering intuition for why scaling may replace traditional TD bootstrapping.
> >
> > While we agree that this would be a valid contribution, it falls outside the scope of our empirical work.

---

### Official Review · Reviewer_TCQQ · 2025-10-31

**Soundness:** 2
**Presentation:** 3
**Contribution:** 2
**Rating:** 4
**Confidence:** 3

**Summary:**

This paper investigates the ability of RL algorithms to compose previously observed fragments of behavior to achieve longer-horizon goals, known as experience stitching. To explore this, the authors introduce a controllable benchmark using the Sokoban game, where agents are trained to perform simple object manipulation tasks and then tested on more complex manipulation tasks that require the composition (or stitching) of previously learned skills.

Building upon this benchmark, the paper presents a series of experiments comparing several Temporal Difference (TD) and Monte Carlo (MC) based RL algorithms. Several key findings are reported: first, MC methods can stitch experiences as well as TD-based methods when the evaluation trajectories lie within the support of the training data. Second, increasing the capacity of the value function allows both TD and MC methods to better generalize when evaluating their ability to compose skills.

The paper has some good ideas such as the taxonomy of Stitching and the testbed to measure Stitching generalization. It has a clear presentation and cohesive writing. But it suffers from poor empirical practices and mischaracterizing and missing prior work. For these reasons I recommend rejecting the paper and I believe with some rewriting and rerunning experiments more carefully, it can be a good paper.

**Strengths:**

* This paper explains and discusses Stitching as a specific way RL algorithms can generalize to new problems. This investigation is novel and interesting to the goal-conditioned RL and Skill and Options communities.
* The writing is coherent and easy to follow, with good use of diagrams to discuss the details of experiments.
*  The introduced testbed is clearly defined, and its design choices and details are explained and justified.
* Discussions around different variants of Stitching and how each experiment is set up to study each variant are coherent and clear.
* The paper mostly manages to place its findings in the context of prior work and justify its existence (with a couple of exceptions, see Weaknesses section)

**Weaknesses:**

* Experiments suffer from several poor empirical practices, which do not allow the reader to fully evaluate the findings.
  * Untuned hyperparameters: all algorithms share the same hyperparameters, and there is no description of how they are selected. This leads me to believe that they are not tuned for this new problem. The performance of untuned algorithms can vary greatly on new problems (Patterson et al., 2023). This fact alone is a major obstacle to trusting the outcome of experiments.
  * The authors report the Inter-Quartile Mean (IQM) of 5 seeds as the performance measure. IQM is not an appropriate choice for this experiment setup because it further obscures the data and shows an average over ~3 seeds, excluding the best and worst performing trials. IQM was designed to aggregate over many problems where the range of returns can vary greatly (e.g., Atari games), not a single problem with a consistent success measure. A better choice would have been to show individual trials if the compute budget only permits 5 seeds.
* Some experiment details are missing or not justified.
  * It is unclear if the evaluation is periodic or only at the end of training. If more than one evaluation is taking place, how was the final number being reported calculated?
  * The experiment is run for 500 million steps, which is a very large number. Why was this number chosen? If it is possible to significantly reduce this number, the rest of the computing budget could have been used to run more trials, which would have made the results more trustworthy.
  *  The appendix reveals the authors' use of a parallel environment setup, but this is not mentioned in the main body. This is an important decision and should be clearly pointed out.
* The authors, on several occasions, fail to define terms that should have been (e.g., waypoint, critic) or make claims about the RL community’s collective beliefs that I do not agree with (e.g., MC methods can not stitch together experiences).
* There is some missing literature that should be included for a complete treatment of the field. There is no mention of the Options framework (e.g., Sutton et al., 1999) or Skill chaining (e.g., Konidaris and Barto, 2009), both are quite related to this paper’s topic.

minor issues,
* Line 106, s and s^\prime should be swapped based on Figure 2
* Lines 108 - 117, labels do not match Figure 2
* Line 247-248, mention by construction twice
* Figure 8: Icons at the (1, 1) coordinates
* different kinds of stitching introduced in the related works section



references
- Patterson, A., Neumann, S., White, M., & White, A. (2023). Empirical Design in Reinforcement Learning. ArXiv, abs/2304.01315.
- Konidaris, G.D., & Barto, A.G. (2009). Skill Discovery in Continuous Reinforcement Learning Domains using Skill Chaining. Neural Information Processing Systems.
- Sutton, R.S., Precup, D., & Singh, S. (1999). Between MDPs and Semi-MDPs: A Framework for Temporal Abstraction in Reinforcement Learning. Artif. Intell., 112, 181-211.

**Questions:**

1. Why was IQM chosen as the reported metric?
2. How are the hyper parameters chosen? and why this choice is justified given that experiments are in a new problem where the algorithms were not tuned for?
3. Line 107, why Sutton Barto cited here? The book does not really cover such an experiment setup
4. Line 319, why was the TD paper cited here? Is there a part of this paper that suggests TD methods can compose sub-behaviours whereas MC methods can not?
5. What is the source of the common wisdom that MC methods can not Stitch experiences together while MC methods can?

---

> ### Author Response · Authors · 2025-11-21
>
> We thank the reviewer for the time they dedicated to reviewing our manuscript. It appears that the reviewer's primary concern is the empirical practices we follow in the paper. Below, we provide detailed information and new experiments that address this concern.
>
> Weaknesses:
>
> > Untuned hyperparameters: all algorithms share the same hyperparameters, and there is no description of how they are selected. This leads me to believe that they are not tuned for this new problem.
>
> In the experimental section, we use the default hyperparameters from OGBench [1] for CRL and GCIQL. For DQN and C-Learning, we keep the same hyperparameters, as they performed well and allowed us to maintain consistent and comparable settings across all methods. To further investigate the impact of hyperparameters, we have added a hyperparameter analysis to the appendix for DQN and CRL, specifically focusing on the discount factor and target entropy that tune the exploration (see Sec. C.2). We use the Quarters Setting with a grid size of 6x6 and 3 boxes, as this setup yielded results that were neither saturated nor trivial for both DQN and CRL.
>
> **Discount factor.** For CRL, the discount factor of 0.9 yields slightly higher training performance than 0.99 for the training task; however, all values of the discount factor produce a trivial policy on the test task. For DQN, the discount factor of 0.99 clearly performs best.
>
> **Target entropy.** For both DQN and CRL, using the target entropy of ln(|A|/2) ≈ 1.1 yields the best results in both training and evaluation tasks. We additionally inspect argmax policies for both CRL and DQN in this setup. Here, again target entropy of ln(|A|/2) ≈ 1.1 yields the highest performance.
>
> We hope this additional analysis addresses the reviewer's concern about the hyperparameters used. If there are other hyperparameters the reviewer thinks should be verified, we are happy to add the additional analysis.
>
> > The authors report the Inter-Quartile Mean (IQM) of 5 seeds as the performance measure. IQM is not an appropriate choice for this experiment setup because it further obscures the data and shows an average over ~3 seeds, excluding the best and worst performing trials.
>
> We agree that IQM may not be the most reliable metric when results are based on only five seeds. To address this, we conducted additional experiments using five more seeds for each setting to improve robustness. The revised manuscript includes updated figures reflecting these results.
>
> > It is unclear if the evaluation is periodic or only at the end of training. If more than one evaluation is taking place, how was the final number being reported calculated?
>
> We evaluate the agents periodically during training; however, the reported results are based on the final five epochs, where the methods have typically converged. Because evaluation also uses parallel environments, we first compute the average success rate for each seed across all evaluation environments, then calculate the average across 5 epochs, and finally compute the IQM across seeds.

---

> > ### Author Response · Authors · 2025-11-21
> >
> > > The experiment is run for 500 million steps, which is a very large number. Why was this number chosen? If it is possible to significantly reduce this number, the rest of the computing budget could have been used to run more trials, which would have made the results more trustworthy.
> >
> > We collect 500 million environment steps and train each method for 5 million gradient updates to ensure that most configurations have enough compute to converge. This data- and update-rich regime prevents us from drawing conclusions based on intermediate performance, which might underestimate methods that converge more slowly. Due to environment parallelization, the data collection process is cheap, and the main bottleneck is the number of gradient updates. As mentioned above, we increased the number of seeds used in every experiment from 5 to 10 in the rebuttal revision of the paper.
> >
> > > The appendix reveals the authors' use of a parallel environment setup, but this is not mentioned in the main body. This is an important decision and should be clearly pointed out.
> >
> > We thank the reviewer for pointing this out and have revised section 5.1 (Experimental Setup) to include that information.
> >
> > > The authors, on several occasions, fail to define terms that should have been (e.g., waypoint, critic)
> >
> > We have added the waypoint and critic definitions in the revised version. Are there any other definitions we may have overlooked?
> >
> > > The authors, on several occasions, … make claims about the RL community’s collective beliefs that I do not agree with (e.g., MC methods can not stitch together experiences).
> >
> > We understand that the claims about conventional beliefs are not universally shared within the community. While various TD-learning-based methods argue that they provide stitching [1,2,3,4], it is much less common in methods based on MC [5,6,7,8]. If the reviewer suggests that we modify the narration in this part, we can incorporate the changes into the new version.
> >
> > > There is some missing literature that should be included for a complete treatment of the field. There is no mention of the Options framework (e.g., Sutton et al., 1999) or Skill chaining (e.g., Konidaris and Barto, 2009), both are quite related to this paper’s topic.
> >
> > Could the reviewer clarify why these two papers should be included in the Related Works section? We are not entirely certain about their direct relevance to our work.

---

> > > ### Author Response · Authors · 2025-11-21
> > >
> > > Questions:
> > >
> > > > Why was IQM chosen as the reported metric?
> > >
> > > We report the IQM metric, as some methods (C-learning, DQN MC) exhibited significant variance across runs. For this reason, we use the metric that is less affected by extreme outliers than the mean. In the revised manuscript, we increased the number of seeds from 5 to 10 to enhance the robustness of the IQM results.
> > >
> > > > How are the hyper parameters chosen? and why this choice is justified given that experiments are in a new problem where the algorithms were not tuned for?
> > >
> > > Please refer to our answer above (in the "weaknesses" comment).
> > >
> > > > Line 319, why was the TD paper cited here? Is there a part of this paper that suggests TD methods can compose sub-behaviours whereas MC methods can not?
> > >
> > > We updated the mentioned sentence to the following form:
> > > Previous works (Ghugare et al., 2024; Myers et al., 2025; Sutton, 1988) argue that temporal difference (TD) methods can compose test-time behavior from sub-behaviors learned during training. However, Monte Carlo (MC) methods might not provide this guarantee.
> > >
> > > > What is the source of the common wisdom that MC methods can not Stitch experiences together while MC methods can?
> > >
> > > Please refer to our answer above (in the "weaknesses" comment).
> > >
> > > References:
> > > [1] Emmons, S., Eysenbach, B., Kostrikov, I., & Levine, S. (2022, May 11). RvS: What is Essential for Offline RL via Supervised Learning? http://arxiv.org/abs/2112.10751
> > > [2] Zheng, C., Salakhutdinov, R., & Eysenbach, B. (2023, October 30). Contrastive Difference Predictive Coding. http://arxiv.org/abs/2310.20141
> > > [3] Kim, J., Lee, S., Kim, W., & Sung, Y. (n.d.). Adaptive Q-Aid for Conditional Supervised Learning in Offline Reinforcement Learning.
> > > [4] Yamagata, T., Khalil, A., & Santos-Rodriguez, R. (2023). Q-learning Decision Transformer: Leveraging Dynamic Programming for Conditional Sequence Modelling in Offline RL. Proceedings of the 40th International Conference on Machine Learning, 38989–39007. https://proceedings.mlr.press/v202/yamagata23a.html
> > > [5] Ghugare, R., Geist, M., Berseth, G., & Eysenbach, B. (2024, January 20). Closing the Gap between TD Learning and Supervised Learning – A Generalisation Point of View. http://arxiv.org/abs/2401.11237
> > > [6] Brandfonbrener, D., Bietti, A., Buckman, J., Laroche, R., & Bruna, J. (2023, January 11). When does return-conditioned supervised learning work for offline reinforcement learning? http://arxiv.org/abs/2206.01079
> > > [7] Kumar, A., Hong, J., Singh, A., & Levine, S. (2022, April 12). When Should We Prefer Offline Reinforcement Learning Over Behavioral Cloning? http://arxiv.org/abs/2204.05618
> > > [8] Janner, M., Li, Q., & Levine, S. (2021, November 29). Offline Reinforcement Learning as One Big Sequence Modeling Problem. http://arxiv.org/abs/2106.02039

---

> ### Comment · Area_Chair_oznk · 2025-11-26
>
> The evaluation of this submission is around the borderline so we need to gather more information from both the authors and the reviewers. Please take a look at the author's response and see how it affects your evaluation.
>
> Best,
> AC

---

> ### Author Response · Authors · 2025-11-26
>
> We appreciate the reviewer’s constructive feedback and are encouraged by their comment that “with some rewriting and rerunning experiments more carefully, it can be a good paper.” We hope that the additional experiments and revisions completed during the rebuttal period adequately address their concerns.
>
> In particular, we updated the appendix (see Section C.2) to include detailed information on how the hyperparameters were selected. In the original submission, we optimized the performance of baselines by running preliminary experiments to tune the following hyperparameters: batch size, number of parallel environments, and number of gradient updates. In addition, during the rebuttal, we conducted further hyperparameter analyses of baselines. One interesting finding: the MC versions of GCDQN, C-learning, and GCIQL perform better with smaller values of the discount factor, likely because lower values of $\gamma$ reduce the variance of relabeled returns [1, 2, 3]. These results reinforce our original conclusions:
> 1. MC methods frequently perform stitching in the generalized setup (see CRL, GCDQN MC, and GCIQL MC in Fig. 10)
> 2. Increasing scale generally improves both training and evaluation success rates for MC and TD methods (see Fig. 8)
>
> We have updated the paper accordingly, revising the figures to include the MC variants of GCIQL, DQN, and C-learning using their best-performing hyperparameters.
>
> References:
>
> [1] Rathnam, S., Parbhoo, S., Swaroop, S., Pan, W., Murphy, S. A., & Doshi-Velez, F. (n.d.). Rethinking Discount Regularization: New Interpretations, Unintended Consequences, and Solutions for Regularization in Reinforcement Learning.
>
> [2] Amit, R., Meir, R., & Ciosek, K. (2020, July 4). Discount Factor as a Regularizer in Reinforcement Learning. http://arxiv.org/abs/2007.02040
>
> [3]  Seijen, H. van, Fatemi, M., & Tavakoli, A. (2019, December 23). Using a Logarithmic Mapping to Enable Lower Discount Factors in Reinforcement Learning. http://arxiv.org/abs/1906.00572

---

### Official Review · Reviewer_bvPx · 2025-11-01

**Soundness:** 2
**Presentation:** 3
**Contribution:** 3
**Rating:** 6
**Confidence:** 3

**Summary:**

This paper compares MC-method and TD-learning in terms of experience sticthing. The authors first characterize a three different stitching scheme : no-stitching (where no end-to end pairs are present in the training set), and exact-stitching  where train set only contains way points, generalized stitching where there are two different way points. Through experiments in a simplified environment, the authors show that MC-based methods can also achieve similar capabilities to that of TD-methods when the critic network capacity is sufficiently large

**Strengths:**

1. The authors tackle an importanat probelm of stitching performance of MC methods and TD methods. The argument that when the critic size is large, there is no significant gap in stitching performance of TD and MC methods is new and will be helpful to the community.

2. The authors try to formalize the stitching concept and the experiments are constructed in solid sense. The designed examples clearly represent the three difference scenarios of stitching that the authors consider.

**Weaknesses:**

1. It is not clear why three different scenarios ( exact stitching, no stitching, generalized stitching) could be representative scenarios to evaluate stitching performance.

2. Structure of presentation : The related works section and preliminaries section seems to be somewhat not balanced : preliminaries overlap with related works making the preliminaries part too short. The authors could provide more detail on their setting. For example, the replay buffer $\mathcal{D}$ is loosely defined, and it is unclear whether it consists of sequence of trajectory or random i.i.d. pairs.

3. Definition of open and closed evaluation presented on page 5: The authors could have adopted a more formal mathematical framework to define these concepts, rather than relying solely on a literal or descriptive explanation.

**Questions:**

1. In the generalized stitching case, do we have $w\to w^{\prime}$ in the training set?


2. What is $\mathrm{supp}(\mathcal{D})$? What do the authors mean by the state support?


3. In Figure 4, when there aer four boxes, the success rate of DQN is than 0.5. Can we consider this as a successful DQN model which we try to do evaluation?

---

> ### Author Response · Authors · 2025-11-21
>
> We thank the reviewer for their time and for reviewing our manuscript. Below, we address the mentioned questions and weaknesses.
>
> Weaknesses:
>
> > It is not clear why three different scenarios ( exact stitching, no stitching, generalized stitching) could be representative scenarios to evaluate stitching performance
>
> We agree that, in principle, many stitching patterns are possible. In this work, we focus on three cases because they form a minimal yet representative ladder of difficulty that aligns with both the literature and our concrete grid setups.
>
> These three regimes are not meant to exhaust all possible stitching behaviors but to cover the main qualitatively distinct cases we can cleanly control in our environment: (i) no recomposition, (ii) stitching with a shared waypoint, and (iii) stitching across mismatched halves. More complex scenarios (longer chains, multiple waypoints, and horizon extension) can be viewed as iterations or combinations of these building blocks, so we consider them representative for evaluating stitching performance in a controlled setting.
>
> > Structure of presentation : The related works section and preliminaries section seems to be somewhat not balanced : preliminaries overlap with related works making the preliminaries part too short. The authors could provide more detail on their setting. For example, the replay buffer  is loosely defined, and it is unclear whether it consists of sequence of trajectory or random i.i.d. pairs
>
> We added clarification about the replay buffer in the revised version of the manuscript. Please let us know if any remaining aspects appear unclear or require further elaboration.
>
> > Definition of open and closed evaluation presented on page 5: The authors could have adopted a more formal mathematical framework to define these concepts, rather than relying solely on a literal or descriptive explanation.
>
> We revised this section and further formalized these concepts. Please let us know if any remaining aspects appear unclear or require further elaboration.

---

> > ### Author Response · Authors · 2025-11-21
> >
> > Questions:
> >
> > > In the generalized stitching case, do we have w->w’  in the training set?
> >
> > We do not assume anything about waypoint-to-waypoint segments w -> w' in D. Our definition of generalized stitching depends only on whether, for the evaluated pair (s,g), there exists a single waypoint w such that both halves (s -> w) and (w -> g) are present; if no such w exists, it's generalized stitching—regardless of any w>w' segments.
> >
> > In our Few-to-Many setup. Training always begins with m > 0 boxes placed, so the buffer lacks any left-half from the zero-placed start used at test. We do not forbid "un-placing," so w->w' that reduce progress could appear, but they are unlikely.
> >
> > > What is D_{supp}? What do the authors mean by the state support?
> >
> > By “state support” D_supp, we mean the part of the state space covered by the training data–generation process. Conceptually, there is a training state distribution P_train(s), and its support is the set of states that occur with non-zero probability. In practice, we approximate this by the set of states that actually appear in the replay buffer, M = {s : s appears in D }, and we refer to M as the (empirical) state support. “Open” stitching then means that solving the test query naturally visits states outside this empirical support, so performance depends on both stitching and robustness to such effectively out-of-distribution states.
> >
> > > In Figure 4, when there aer four boxes, the success rate of DQN is than 0.5. Can we consider this as a successful DQN model which we try to do evaluation?
> >
> > In Figure 4, the DQN success rate for the four boxes is approximately 35% on the training task and nearly 0% performance on the test task (diagonal movement). We consider this an unsuccessful performance, and evidence that for more complex tasks, stitching is harder, as the gap between training and test performance increases with the number of boxes present. This result indicates that even the TD method (DQN) cannot stitch together well the behaviors present in the training experience.

---

### Official Review · Reviewer_bjbE · 2025-11-01

**Soundness:** 2
**Presentation:** 3
**Contribution:** 3
**Rating:** 6
**Confidence:** 4

**Summary:**

This paper considers the ability for algorithms to stitch segments of experience together, something typically seen as a key property of temporal difference methods as they explicitly encourage this through bootstrapping. The authors note that with complex feature spaces, the odds of paths intersecting is vanishingly small that the resulting stitching is enabled through generalization between nearby states. They use this to motivate a systematic study of stitching, and empirically demonstrate that Monte Carlo returns—an extreme which does no explicit stitching—can solve problems that require stitching. They further show that scaling the size of the network generally improves stitching ability, suggesting that techniques which promote generalization within neural networks are more important for the ability to stitch than temporal difference updates.

**Strengths:**

* The paper is well-motivated. While TD is often contrasted with MC by this ability to stitch, such intuitions are presented in tabular settings. The observation that generalization is necessary for this to happen with function approximation is lesser discussed.

* The paper provides a useful classification of stitching regimes: no stitching, exact stitching, and generalized stitching.

* They further detail an environment setup and how start/goal states can be configured to systematically test for each stitching regime with function approximation. Through this controlled experimental setup, they highlight two key observations: MC can sometimes stitch, and that scale is a key enabler of stitching.

**Weaknesses:**

* The evidence of MC stitching was notable in the generalized stitching regime, but surprisingly significantly less prevalent in the exact stitching case. This seems weird, if the setup which cleanly sets up trajectories for stitching led to worse stitching ability?

* Assuming GCDQN (MC) is a sound algorithm, there's no comparison between GCDQN (TD) and GCDQN (MC) in the comparisons of Section 5.2, despite it being used in 5.3. It feels like it would be a fairer and more convincing comparison between TD and MC if the base algorithm can be controlled. It's hard to draw conclusions between TD vs. MC if the results overall show substantial variability by underlying method (e.g., CRL and C-LEARN are both MC yet perform dramatically differently).

* Only 5 seeds were used which has been repeatedly shown to be questionable with regard to making proper statistical comparisons for the claims being made (e.g., Henderson et al., 2017; Colas et al., 2018; Patterson et al, 2023; Patterson et al, 2024). It would strengthen the paper to justify the number of seeds (e.g., by observing the trends in success rate over task difficulty, making it a larger number of seeds over a distribution of tasks, etc.).

**Questions:**

* The low performance of DQN MC was noted to potentially be an exploration issue—can the authors elaborate on why they believe this to be the case?

* On another note, how exactly was DQN MC implemented? It seems to be at odds with many aspects of DQN (e.g., replay buffers, off-policy bootstrapping without importance sampling, etc.).

* The title and tone of the paper gives the impression that in addition to challenging an idea that TD is uniquely capable of stitching, the other factors might be more important to the ability to stitch than the use of TD updates. e.g., by questioning whether it's the "gold standard", highlighting that the gap between small to large networks is larger than the gap between TD and MC, suggesting that TD is "less necessary", etc. However, the results still seem highly favorable for TD, where in all cases that MC exhibited stitching, TD stitched significantly better. The extent of the benefit from scale seemed highly variable depending on the algorithm choice, with the largest improvement through scale still being a TD method. It's unclear whether the title and tone are warranted, in light of this—can the authors comment on the choice to frame the exposition in this way?

---

> ### Author Response · Authors · 2025-11-21
>
> We thank the reviewer for their time and for reviewing our manuscript. Below, we address the mentioned questions and weaknesses.
>
> Weaknesses:
>
> > The evidence of MC stitching was notable in the generalized stitching regime, but surprisingly significantly less prevalent in the exact stitching case. This seems weird, if the setup that cleanly sets up trajectories for stitching led to worse stitching ability?
>
> We agree! When building the experimental setup, we expected the two problem settings (Quadrants and Few-to-many) to yield similar results. To our surprise, the Quadrants setting was found to be much more challenging for stitching. Through controlled experiments and analysis (see Sec. 5.2, especially Fig. 5), we probed why this problem was more challenging. We conjecture that the underlying reason is that in the exact stitching (Quadrants) task, during test-time, most methods, especially MC ones, attempt to solve it by moving boxes directly to the diagonal quadrant. This induces out-of-distribution states when more than one box is present on the board (see Open setup definition Sec. 4). As the number of boxes increases, the likelihood of encountering such OOD states grows accordingly.
>
>
> > Assuming GCDQN (MC) is a sound algorithm, there's no comparison between GCDQN (TD) and GCDQN (MC) in the comparisons of Section 5.2, despite it being used in 5.3. It feels like it would be a fairer and more convincing comparison between TD and MC if the base algorithm can be controlled. It's hard to draw conclusions between TD vs. MC if the results overall show substantial variability by underlying method (e.g., CRL and C-LEARN are both MC yet perform dramatically differently).
>
> We plan to add the missing comparison of GCDQN (TD) and GCDQN (MC) to the appendix next week.
> In the comparisons in Section 5.2, we decided to include only the implementations of representative MC and TD methods in their original forms, rather than their modifications.
> We note that comparing the performance between different methods, for instance, GCDQN (TD) to CRL (MC), is not the goal of the paper. We rather present these methods as instantiations of the particular learning methodology (TD or MC) to analyze their stitching capabilities. For this reason, we implemented different versions of GCDQN, C-learning, and GCIQL, based on MC and TD targets.
>
>  > Only 5 seeds were used which has been repeatedly shown to be questionable with regard to making proper statistical comparisons for the claims being made …
>
> To address this, we conducted additional experiments using five more seeds for each setting to improve robustness. The revised manuscript includes updated figures reflecting these results.

---

> ### Author Response · Authors · 2025-11-21
>
> Questions:
>
> > The low performance of DQN MC was noted to potentially be an exploration issue—can the authors elaborate on why they believe this to be the case?
>
> We believe that exploration is one of several factors contributing to the low performance of DQN MC. In particular, the variance of relabeled returns collected during exploration depends heavily on the agent’s behavior after reaching the goal. If the agent reaches the relabeled goal at step $k$ and then keeps the box configuration unchanged until the episode ends at step $T$, the computed return will be high, proportional to $T - k$, because the reward is defined as a simple equality check between the next-state configuration and the goal. However, if the agent reaches the goal at step $k$ but later alters the configuration at step $t<T$, the return drops to being proportional only to $t - k$. We hypothesize that a lower discount factor may help reduce this variance, albeit at the cost of shortening the effective horizon. We are currently running experiments to test this idea.
>
> We also observe that smaller DQN MC architectures learn significantly slower than larger ones. Another contributing factor may be that MC returns are inherently defined for on-policy data, while our implementation relies on a large replay buffer, which could introduce data staleness. Finally, both DQN MC and DQN TD use a 50/50 mixture of future states and random states as HER goals. In TD, random-goal targets are often non-zero due to bootstrapping, whereas in DQN MC they are typically zero, which may further slow learning.
>
> > On another note, how exactly was DQN MC implemented? It seems to be at odds with many aspects of DQN (e.g., replay buffers, off-policy bootstrapping without importance sampling, etc.).
>
> We acknowledge that referring to this method as DQN (MC) may have been a suboptimal naming choice, as it does not closely align with the original DQN algorithm described in [1]. We selected this name to highlight the primary difference, namely, how the targets are computed. DQN (MC) is implemented by changing the bootstraped DQN (TD) target to the actual cumulative reward received in an episode. We implemented a version of the trajectory replay buffer that utilizes experience relabeling across the entire trajectory. This means that we swap goals throughout the entire trajectory stored in the replay buffer to match the state achieved in the future in the same trajectory, sampled using a geometrical distribution. We then compute MC rewards by going ‘backwards’ through the trajectory and applying a discount to the reward at each step. The goal of this algorithm was to investigate how performance and generalization are specifically tied to TD-style rewards, and whether similar outcomes can be achieved using MC-style rewards. We have clarified this implementation detail in section 5.1.
>
> > The title and tone of the paper gives the impression that in addition to challenging an idea that TD is uniquely capable of stitching, the other factors might be more important to the ability to stitch than the use of TD updates. e.g., by questioning whether it's the "gold standard", highlighting that the gap between small to large networks is larger than the gap between TD and MC, suggesting that TD is "less necessary", etc.
>
> We agree that the scaling improves both MC and TD methods in our setup. We also acknowledge that the gap between small to large networks is not larger than the gap between TD and MC. Accordingly, we have revised the relevant claims in the paper, as well as the title (“Is Temporal-Difference Learning the Only Path to Stitching in RL?”). We will update it at the earliest opportunity.
>
> Nonetheless, we believe that our paper makes important advances by (1) studying the extent to which ``stitching'' actually makes sense in discrete settings with function approximation (definitions, benchmark, and extensive analysis in Sec. 5), (2) finding empirically that model depth might play an important role in stitching (improving both TD and MC methods performance as shown in Sec. 5.4), and (3) finding that Monte Carlo methods can exhibit a stitching property that several prior works argue they should be unable to display [2,3,4,5] (as demonstrated in Sec. 5.3).

---

> ### Author Response · Authors · 2025-11-21
>
> References:
>
> [1] Mnih, V., Kavukcuoglu, K., Silver, D., Graves, A., Antonoglou, I., Wierstra, D., & Riedmiller, M. (2013, December 19). Playing Atari with Deep Reinforcement Learning. http://arxiv.org/abs/1312.5602
>
> [2] Ghugare, R., Geist, M., Berseth, G., & Eysenbach, B. (2024, January 20). Closing the Gap between TD Learning and Supervised Learning – A Generalisation Point of View. http://arxiv.org/abs/2401.11237
>
> [3] Brandfonbrener, D., Bietti, A., Buckman, J., Laroche, R., & Bruna, J. (2023, January 11). When does return-conditioned supervised learning work for offline reinforcement learning? http://arxiv.org/abs/2206.01079
>
> [4] Kumar, A., Hong, J., Singh, A., & Levine, S. (2022, April 12). When Should We Prefer Offline Reinforcement Learning Over Behavioral Cloning? http://arxiv.org/abs/2204.05618
>
> [5] Janner, M., Li, Q., & Levine, S. (2021, November 29). Offline Reinforcement Learning as One Big Sequence Modeling Problem. http://arxiv.org/abs/2106.02039

---

### Author Response · Authors · 2025-11-21

We thank the reviewers for the time and care they devoted to evaluating our manuscript. Their insights have helped us further improve both its quality and readability. In the revised version, we have primarily:
* Provided an analysis of hyperparameters used for DQN and CRL (see Sec. C.2) and investigated argmax policy induced by Q function (see Sec. C.3).
* Increased the number of seeds in all experiments from 5 to 10.
* Updated the text to incorporate the reviewers’ suggestions.
* We also added a goal-conditioned version of Implicit Q-Learning, both the TD and MC versions.

We have marked all textual changes in blue and highlighted the updated plots (improved visualization or additional seeds) using blue captions.

We believe these revisions significantly improve the quality of our manuscript, and we are once again grateful to the reviewers for their helpful suggestions.

---

> ### Author Response · Authors · 2025-11-30
>
> Dear Area Chair and Reviewers,
>
> Here's a summary of the key contributions of the paper, the key points raised by the reviewers, and the changes made to the paper to address these concerns. Please let us know if there's any other information we can provide!
>
> Our work investigates whether temporal-difference (TD) learning is the unique path to _experience stitching_ in reinforcement learning. To study this question, we first introduce a discrete, goal-conditioned Sokoban-style benchmark that tests a few different types of stitching. Our primary aim is to fact-check the oft-cited claim that Monte Carlo (MC) methods cannot perform stitching. Our primary finding is that MC methods can perform stitching. Our experiments highlight that this is particularly true for certain types of stitching and when value functions are sufficiently large. Taken together, our empirical results and benchmark provide a new mental model for understanding stitching, an important yet elusive property for RL algorithms.
>
> The submission received three borderline-accept ratings and one borderline-reject (scores: 6, 6, 6, 4). While the reviewers praised the careful experimental design and the proposed taxonomy of stitching, concerns were raised regarding empirical rigor, scope, and definitions. Reviewers bjbE and TCQQ argued that 5 random seeds were statistically insufficient, with Reviewer TCQQ noting that the Inter-Quartile Mean (IQM) is an inappropriate metric for such a small sample size. Regarding the experimental scope, Reviewer 18Fp commented that the focus on discrete grid-worlds is restricted and suggested validation in continuous control environments. On the algorithmic front, Reviewer bjbE noted the absence of a direct comparison between TD and MC versions of GCDQN and inquired about the implementation details of "DQN (MC)", while Reviewer TCQQ raised concerns about using untuned hyperparameters across different algorithms. Finally, Reviewer bvPx requested more formal mathematical definitions for "open" and "closed" evaluation settings and inquired about the rationale behind choosing the three specific stitching scenarios as representative. As detailed below, we have revised the paper to run the additional experiments and analyses suggested by the reviewers, and these experiments continue to support the main conclusions from the paper.
>
> We summarize the main clarifications and actions we have taken to address concerns during the rebuttal period:
>
> - **Reviewers bjbE and TCQQ** raised concerns regarding statistical robustness with only 5 seeds. In response, we have **increased the number of seeds from 5 to 10** for all experiments in the revised manuscript to ensure our claims are statistically sound.
> - **Reviewer TCQQ** requested more transparency regarding hyperparameter selection. We have added a detailed description of hyperparameter selection process and **a new analysis in Appendix C.2**, specifically investigating the discount factor and target entropy for DQN and CRL. Additionally, **in Appendix C.4**, we include an analysis of how different values of discount interact with different scales of the critic network.
> - **Reviewer bjbE** noted the absence of a direct comparison between GCDQN (TD) and GCDQN (MC). We clarify that this comparison is already provided in **Appendix C.1**, and we have further expanded the analysis **in Appendix C.4**. The reviewer also inquired about the details of GCDQN (MC) implementation, which we have **added to Sec 5.1**.
> - **Reviewers bvPx and TCQQ** noted that several concepts, such as waypoints, the critic, and replay buffer details, were insufficiently defined. We have revised **Sections 3, 4, 5**, as well as the relevant definitions, to provide precise formalism and ensure the experimental setup is fully transparent.
> - **Reviewer 18Fp** requested an analysis of exploration efficiency. In response, we added experiments **in Section C.3** examining how the temperature of the softmax Q-induced data collection policy influences exploration and, in turn, affects stitching.
> - **Reviewer 18Fp** noted that the focus on discrete grid-worlds might be too restrictive. In our limitations, we explain that we intentionally chose this controlled setup to enable a concrete evaluation of stitching, which is challenging to verify in more complex domains.
> - **Reviewer bjbE** commented on the tone of the title. We have revised the title to **“Is Temporal-Difference Learning the Only Path to Stitching in RL?”** to better reflect the nuance of our findings.
>
> We believe these revisions significantly improve the quality and robustness of our manuscript. We remain available for any further questions.

---

### Meta-Review · Area_Chair_QkqN · 2025-12-24

**Summary:**

This paper performs an empirical study on the experience stitching ability of TD and MC methods (with function approximation) in large RL problems. Through careful experimental design, the authors claim that their observation reveals the fact that MC also has the stitching ability (which was believed not an ability of MC). They also observe that, in line with traditional wisdom, TD has stronger ability of experience stitching than MC.

The overall rating of the paper is borderline. During the discussion period, the authors have addressed most of the concerns from the reviewers. But there still remain several criticisms that have yet been clarified in a satisfactory way. These issues include (1) Reviewer TCQQ reports concerns on parameter tuning issues, though the authors have made some justification, yet the explanations are not convincing enough; (2) This reviewer also reports issue on the use of IQM. The authors add 5 seeds to the experiment but, still, not using a more ``transparent'' way that the reviewer expects; (3) About stitching in TD \& MC. The narrative is bit improper. The paper does not disprove the stitching ability of TD, in fact TD still performs better than MC, so the current title is not appropriate (the authors promise to revise this). However, the community's ``common belief that MC does not have stitching ability'' is also not well-supported by the authors arguments. So, the overall narrative of the paper needs a major revision. That is, in terms of stitching, TD is still good, while MC is not widely believed to be bad.

Based on this discussion. The AC thinks that the rebuttal is not sufficient to change this reviewer's overall opinion. And the AC also thinks these concerns are meaningful. Therefore, the AC decide to reject this paper. (The AC still believes in the value of this paper, and the AC encourages the paper to be resubmitted to a good venue after fully addressing these issues)

**Reviewer Concerns:**

Addressed:
(1) bjbE: Lack of comparison between GCDQN (TD) and GCDQN (MC).
(2) bjbE & TCQQ: Limited number of seeds in the experiments.
(3) bvPx: Several presentation issues.

Not Addressed:
(1) TCQQ: The choice of IQM as performance metric (the authors provide some justification but not fully convincing).
(2) TCQQ: The untuned hyperparameters: all algorithms share the same hyperparameters, and there is no description of how they are selected (the authors provide some justification but not fully convincing).
(3) TCQQ: The positioning of the paper.

**Reviewer Scores:**

The main negative comment comes from reviewer TCQQ. Though the authors made some justification, the AC think they are not sufficient to change the reviewer's mind. Therefore, the overall score of the paper will be marginally below the conference's threshold.

---

### Decision · Program_Chairs · 2026-01-26

Reject